# The Imperative of Regulation: The Co-Creation of a Medical and Non-Medical US Opioid Crisis

Toine Pieters [1,2]

1 Freudenthal Institute, Utrecht University, P.O. Box 85 170, 3508 AD Utrecht, The Netherlands; t.pieters@uu.nl
2 Department of Pharmaceutical Sciences, Utrecht Institute for Pharmaceutical Sciences (UIPS), Utrecht University, P.O. Box 80 082, 3508 TB Utrecht, The Netherlands

**Abstract:** The ravaging COVID-19 pandemic has almost pushed into oblivion the fact that the United States is still struggling with an immense addiction crisis. Drug overdose deaths rose from 16,849 in 1999 to nearly 110,000—of which an estimated 75,000 involved opioids—in 2022. On a yearly basis, the opioid casualty rate is higher than the combined number of victims of firearm violence and car accidents. The COVID-19 epidemic might have helped to worsen the addiction crisis by stimulating drug use among adolescents and diverting national attention to yet another public health crisis. In the past decade, the sharpest increase in deaths occurred among those related to fentanyl and fentanyl analogs (illicitly manufactured, synthetic opioids of greater potency). In the first opioid crisis wave (1998–2010), opioid-related deaths were mainly associated with prescription opioids such as Oxycontin (oxycodone hydrochloride). The mass prescription of these narcotic drugs did anything but control the pervasive phenomenon of 'addiction on prescription' that played such an important role in the emergence and robustness of the US opioid crisis. Using a long-term drug lifecycle analytic approach, in this article I will show how opioid-producing pharmaceutical companies created a medical market for opioid painkillers. They thus fueled a consumer demand for potent opioid drugs that was eagerly capitalized on by criminal entrepreneurs and their international logistic networks. I will also point out the failure of US authorities to effectively respond to this crisis due to the gap between narcotic product regulation, regulation of marketing practices and the rise of a corporate-dominated health care system. Ironically, this turned the most powerful geopolitical force in the war against drugs into its greatest victim. Due to formulary availability and regulatory barriers to accessibility, European countries have been relatively protected against following suit the US opioid crisis.

**Keywords:** regulatory science; opioid crisis; drug lifecycle; addiction on prescription; opioids; pharmaceutical company; medical professionals



## 1. Introduction

The ravaging COVID-19 pandemic has almost pushed into oblivion the fact that the United States (US) is still struggling with an immense addiction crisis. Drug overdose deaths rose from 16,849 in 1999 to 109,680—of which about 75,000 involved opioids—in 2022 [1,2]. According to WHO estimates, this amounts to approximately 60% of the worldwide number of people dying of an opioid overdose [3]. For more than a decade, drug overdose has been the leading cause of injury death in the USA. On a yearly basis, the drug overdose mortality rate is higher than the combined number of victims of firearm violence and car accidents [4]. The 2020 and 2021 data indicated that the US opioid epidemic showed a significant upsurge in the number of drug overdose deaths [5]. In 2022, the number of overdose deaths seemed to have leveled off [6]. According to recent research, the COVID-19 crisis might have helped to worsen the addiction crisis by stimulating drug use among adolescents and diverting national attention to yet another public health crisis [7,8].

The addiction crisis and the COVID-19 pandemic, widely considered as two consecutive and intersecting public health emergencies, lay bare vulnerabilities and inequities of the US health and social–economic systems [9,10]. As sociologist Paul Starr has shown convincingly in his history of the American health care system, the very system that made possible the bounty of the American high-tech, cure-focused American hospital settings, with their vaunted miracle cures and life-prolonging medical interventions, has also become a symbol of national frustration due to exploding health care costs and a lack of coordination and inconsistent government regulation [11].

In the past two decades, more than a million Americans have lost their lives to the opioid crisis [12]. Drug overdoses are the leading cause of death for Americans under the age of fifty [13]. The sharpest increase in deaths has occurred among those related to fentanyl and fentanyl analogs (illicitly manufactured, synthetic opioids of greater potency) in the period 2016–2022 [14]. The impact is also evident in life expectancy statistics. Though the opioid crisis is not the only reason, as of 2018, the life expectancy of Americans has fallen for the first time in more than three decades [15,16]. While middle-class, non-Hispanic white Americans living in non-urban areas were particularly affected during the first decade of the opioid crisis, the substance use disorder (SUD) problem has crossed every geographic and racial boundary in the past decade [17]. Data show that 8.7 million children in the USA under the age of 17 live in households with a parent who has an SUD [18]. In 2019, the creators of the children's TV show Sesame Street were even spurred into taking on a new, unusual topic to help American children navigate the dark side of life in the USA: the opioid crisis [19].

A great deal of research and evidence has begun to emerge to explain the historical and contemporary factors that have facilitated the emergence and acceleration of this crisis [20–23]. This vast literature makes clear that the crisis is multifaceted and complex, discussing the social, economic and corporate roots of the crisis. So far, the evidence as part of the numerous litigation cases throughout the USA primarily points to the major role of the opioid-producing pharmaceutical companies like Purdue Pharma in fueling the first wave of the opioid epidemic (1998–2010) [24]. However, the changing dynamics of the opioid crisis, the nature of regulatory conditions and, more specifically, the interference between medical and non-medical supply and demand chains has received less scholarly attention.

The concerns of governments across the globe about the persistent problem of the harmful non-medical use of narcotic prescription drugs has resulted in a growing international list of controlled narcotic prescription drugs. The Single Convention on Narcotic Drugs (1961) is considered the bedrock of the current United Nations-based global prohibition-centered drug control regime [25]. The prohibitive framework consists of five different schedules of controlled substances, numbered I–V. The different schedules are based on three factors: potential for abuse, accepted medical use and safety and potential for addiction. Schedule I drugs are considered to have the highest potential for abuse and are not accepted for regular medical use, whereas schedule V drugs have the lowest potential for abuse and are regularly prescribed in medical practice. Schedule II opioid drugs are accepted for medical use but, with a high-risk profile for both abuse and addiction, prescription monitoring is strongly advised (though not necessarily enforced) by authorities in all US states [26]. According to drug historian Stephen Snelders, the very lack of success in effectively controlling substance use has driven a continual expansion of the framework and evoked its own antithesis: the further development and expansion of the illegal drug trade to meet a continuously growing consumer demand [27].

In order to use a therapeutic medicine such as an opiate pain relief medication US citizens must obtain a prescription from a licensed and qualified medical doctor. Yet, before an American physician can prescribe a drug, the US Food and Drug Administration (FDA) must have approved it. In Europe, the European Medicines Agency (EMA) has the same authority. No new drug can be legally marketed in the United States before the FDA (or the EMA in Europe) has formally declared it 'safe and effective' for its intended use. In both Europe and the USA, this requires a five-step process: discovery/concept, preclinical

research, clinical research, FDA/EMA review and FDA/EMA post-market monitoring. After an extensive testing trajectory, pharmaceutical companies send regulators a new drug application (NDA), which includes all the drug's preclinical and clinical test results, all manufacturing information and the proposed label for use of the drug. This submission document is then reviewed by a panel of experts and the drug can be allowed to the market only after an approval decision.

In principle, the FDA/EMA will continue to monitor the drug post-approval and, if needed for safety reasons, can order pharmaceutical producers to adjust the product label or even to alter the manufacturing process or product properties [28,29]. In emergency situations, the drug can even be removed from the market. Historically, FDA's and EMA's primary focus and executive power has been upon pre-marketing product regulation and far less on drug post-approval monitoring [30]. Thus, both FDA and EMA have the same goal: to protect public health by ensuring compliance with the medication's safety, efficacy and quality. However, FDA is a centralized government agency that oversees the drug development and marketing process in a single country (USA), whereas EMA coordinates centralized procedures and national competent authorities in various member states of the European Union (EU). This implies that national laws and regulations may apply at the country level in the EU [31–33]. As I will show, this resulted in rather different opioid regulatory practices and barriers to opioid accessibility in the EU as compared with the USA.

By using a long-term drug lifecycle analytic approach, I will point out how the historically grown gap between narcotic product regulation, regulation of pharmaceutical marketing practices and the rise of a corporate-dominated health care system played an important role in the failure of US authorities to effectively respond to the opioid crisis. Finally, I will provide some cross-comparisons about the use and regulation of opioid medicines in non-US regulatory systems, with a focus on Europe.

## 2. Materials and Methods

For this article, I employed a mixed-method approach involving archival research in Dutch and German early-twentieth-century medical and pharmaceutical journals for tracing the early drug life of oxycodone and a narrative review of the more recent literature. PubMed was searched for relevant publications by using the search terms "*Opioid crisis*" OR "*Opioid epidemic*" OR "*Opioid regulation*". For this Pubmed search, I excluded research before 1990 and articles published in languages other than English. To maximize discovery of eligible articles, the citations and references of included articles and related reviews were investigated ('snowballing').

The drug life cycle analysis in this article was based on the life-cycle concept and methodology developed within my research group from 2005 onwards, building on the drug trajectories work in the field of Science and Technology studies [34–36]. For a better understanding of the subsequent historical dynamics related to the development of new generations of opiates and opioids by the pharmaceutical industry, it is important to take into account the cyclical nature of drug trajectories. There are generally four stages of the drug life cycle. First, there is a testing and approval trajectory. Second, after the drug is introduced, there is market expansion, growing public expectations and multiplication of drug indications. Next, drug maturity with a high sales volume is usually accompanied by rising criticism and disappointment regarding drug effectiveness and side effects. Finally, there is reduced use and drug application becomes more limited. These phases need not be sequential: they often overlap. The cycle often ends with the disappearance of the drug from the medical market and its replacement by newer drugs which will follow similar trajectories. But a drug can also reappear and start a new life cycle. Drug life cycles involve continuous interactions between stakeholders within academia, the healthcare sector, the government and industry within the public sphere (including drug-using subcultures). As a result, a drug's medical and public profile is under constant revision [37,38]. New diagnostic, therapeutic and recreational categories emerge and change as social and material

conditions change. Doctors, patients and other consumers have to learn how to use and cope with new generations of psychoactive substances between the laboratory, the bedside and the household, 'redefining the boundaries between healing and soothing the mind, and fulfilling fashionable desires of comfort, convenience and pleasure' [39]. As I will show, these cyclical historical dynamics seem to play an important role in the recurrent nature of epidemics of pharmaceutical drug abuse in America [40].

## 3. Results

### 3.1. Doctors and Pharmacists as Gatekeepers of Narcotic Drugs

Drug historians David Courtwright and Virginia Berridge have shown how political elites in the USA and Europe in the late nineteenth and early twentieth centuries came to view and define non-medical narcotic use as a problem that governments needed to regulate [41,42]. Authorities on both sides of the Atlantic were struggling with the overconsumption of opiates in the period 1880–1920, but the impact of the opiate crisis was felt most dramatically in the US with an exponential rise in the number of morphine, heroin and cocaine addicts in the 1890s [43]. Under American leadership, diplomats worked on an international drug control regime; starting with the Hague Convention of 1912, they sought through international conferences and treaties to prohibit the non-medical use of narcotics and control the supply chains [44]. They called for stricter regulations to enable maximum legislative control with doctors and druggists as gatekeepers of narcotic drugs [45]. The Harrison Act in the United States (1915) aimed also at limiting the medical use of opiates and thus curbing consumer demand, followed by similar restrictive opiate legislation in other countries. There were penalties for pharmacists, doctors and addicted patients who infringed the laws, and for any party without legal permission importing or exporting narcotics [46]. The new prohibitive international regime would leave a hole in family medicine cabinets in the USA and Europe [39]. Doctors and lay consumers in search of prescription medicines with sedative, hypnotic and pain-relieving properties did not have to wait long before alternative drugs would be available. The burgeoning international pharmaceutical industry started developing new generations of prescription-only chemically synthesized sedatives, hypnotics and painkillers. The foundations were thus laid for the emergence of a new mass market for prescription-only, psycho-active and pain-relieving medicines under the supervision of doctors and pharmacists [47].

### 3.2. Regulating New Addictive Wonders for the Doctor's Bag

The public appetite for more potent, faster-acting and safer hypnotics, sedatives and painkillers continued to grow. German, British, French and American pharmaceutical companies did their utmost to meet these relief demands with ambitious drug screening programs [48]. While the Canadian judge Emily Murphy issued an early warning against the use of these new psychoactive compounds, stating that 'anything that acts like an opiate *IS* an opiate', a new analgesic and cough medicine oxycodeinon (oxycodone) under the tradename Eukodal was launched in 1919 by German pharmaceutical company E. Merck (Darmstadt) [49,50].

The name Eukodal was intentionally chosen to differentiate the drug (in marketing terms) from conventional opiates and to associate it with the family of barbiturates that was still officially considered to be non-habit forming [51]. Doctors were made to believe that this new, more potent pain medicine could be used without special precautions and among other therapeutic uses (analgesic, narcotic, cough medicine) as an antidote against morphinism [52,53]. Eukodal was introduced into a drug market saturated with sedatives, hypnotics and stimulants, which were available in dozens of formulas and brands. The relatively high price of the new prescription drug also limited its use [54]. In the late 1920s and early 1930s, warnings about addiction ('Eukodalism'), intoxication risks and counterfeit prescriptions began to circulate in European medical and pharmaceutical journals [55–57]. In response, the regulatory authorities across Europe and the USA labeled Eukodal as an opiate medicine with restrictive prescription rules [58]. With the expiration of the

oxycodone patent, Eukodal gradually disappeared from the medical market in Europe, surviving only temporarily in the form of unnecessary war supplies that ended up in the criminal narcotic trade [59]. As part of the parallel dynamics of drug development and drug regulatory regimes, we will see a further spiraling of medically controlled licit ('on prescription') drug markets and increasingly criminally controlled illicit drug markets in the post-World War II period.

The international production, distribution and use of opiates was further restricted after the Second World War. Supply control was dominant, with a reduction in the nonmedical markets sought through curtailing and monitoring of excess capacity for the medical markets of opium producers and narcotic drug manufacturers. Regulation to avoid abuse was deemed essential, but overly stringent regulation that prevented patients from pain relief caused them to seek alternative means of obtaining these drugs and that had likewise to be prevented [60] (p. 145).

The driving force behind the tightening of the international prohibition regime and the focus on supply was Harry J. Anslinger, the head of the US Federal Bureau of Narcotics (FBN). He played a crucial role in the US-led war on drugs by simplifying the existing increasingly unwieldy drug control machinery and persuading the United Nations in taking the lead to develop a unifying and uniform international prohibition-oriented treaty. The resulting 1961 United Nations Single Convention on Narcotic Drugs came indeed close to imposing an international prohibition regime for narcotic drugs. Addiction to narcotic drugs was presented in the treaty as a 'serious evil for the individual' that is 'fraught with social and economic danger to mankind' [25]. A key provision of the Convention imposed the obligation to all international parties to take such legislative and administrative measure 'to limit exclusively to medical and scientific purposes the production, manufacture, export, import, distribution of, trade in, and possession of narcotic drugs' [25]. In addition to addressing control issues, the Convention obligated countries also to work towards adequate medical access to narcotic drugs to alleviate pain and suffering, but that would prove rather difficult. As a safeguard to limit the use of narcotics, the international Narcotics Control Board (INCB) was set up to monitor the global production and distribution chains in order to prevent the medical and non-medical supply chains from playing off one another.

The Convention marked a significant change in the national policies regarding legal barriers in accessing opiate medicines. Across the world, amplified by a wave of opiophobia, stricter control measures in national policies and legislation were implemented further impeding access to legitimate medical use of opiate medicines [60] (p. 146). In most European countries, the prescribing of morphine-like controlled substances would require a permit to prescribe or receive opiates, multicopy prescription requirement on special forms and limitations on the treatment period and on the dispensing privileges. The resulting far-reaching control measures would for decades cast a shadow on the adequate access to the steadily growing bag of controlled narcotic medicines for medical and scientific purposes [61]. Underprescription of opiates became the rule in the USA and most other countries [62].

The focus of the supply-side drug control regime was more on non-Western plant-based narcotic drugs and less on the new generations of psychotropic products of the Western chemical and pharmaceutical industries. The latter succeeded in meeting market demands by formally avoiding the international drug control regime in the 1950s and early 1960s. This is exemplified by the industrial product wave of hypnotics (e.g., barbiturates and benzodiazepines), sedatives (e.g., meprobamate) and stimulants (amphetamines) for inducing sleep, soothing nerves and brightening Cold War-infused anxious moods [36,63]. Medical morphine use might have dwindled but addiction on prescription continued in an 'upper' and 'downer' pill disguise [47]. In addition, the over-production and over-supply by the pharmaceutical industry of these medical stimulants and sedatives fed into global illegal markets [39]. In the USA, this would generate a further legislative response in the form of the 1965 Drug Abuse Control Amendments, which brought the

manufacture, distribution and sale of barbiturates, amphetamines and tranquilizers under federal control [64]. Consequentially, the number of controlled drugs under medical supervision of doctors and pharmacists rose exponentially, thus placing a significant additional burden on their task as gatekeepers.

Yet another telling example of this addictive pharmaceutical product diversification is the approval by the FDA in 1950 of a 'new' analgesic, developed by the small US pharmaceutical company Endo Products, consisting of a low dose of oxycodone (4.8 mg) and aspirin (325 mg) under the tradename Percodan. It was presented as a revolution in pain care and fueled a second drug life cycle of oxycodone. The combination of the 'wonderous drug' aspirin with a claimed non-addictive opioid snuff, that did not qualify for a narcotic drug label, was advertised as a quick fix in pain relief and was hailed by doctors and patients alike. Percodan, however, would become an officially recognized target of abuse by the early 1960s, with famous drug users like Marilyn Monroe and Elvis Pressley, prompting a change from a 'class B' narcotic with no prescription obligation to a 'class A' (schedule II) narcotic, which required a written prescription [65]. This did not hold Endo Products back. After being taken over by the US chemical company Dupont and renamed as Endo Laboratories L.L.C., in 1976, the company introduced Percocet (a low-dose oxycodone, acetaminophen combination), which was proclaimed to be a safer fast-relief analgesic alternative to Percodan [66]. Like Percodan, Percocet became a cash cow for Endo and, despite the US-wide prescription-only regulation for both medicines, abuse was widespread [67].

### 3.3. Unforeseen Consequences of a New Dosage Form: Slow Release

Morphine as a palliation medication survived the 'opiaphobia' and 'morphinophobia' which continued to circulate in pain specialist quarters in America and Europe [68–71]. Most specialists in the 1970s advocated for multidisciplinary pain treatment programs involving physical therapy, psychotherapy and additional pharmacological therapy in the form of the new generation of nonnarcotic analgesics (e.g., non-steroidal anti-inflammatory drugs, NSAIDS such as ibuprofen) and psychotropic drugs. An intensified war on drugs during the Nixon era raised the regulatory stakes once again for American doctors prescribing narcotic relief-bringing drugs for their patients [72]. Undertreatment of cancer pain and chronic pain became the rule rather than the exception in the 1970s and 1980s [73]. With the exponential growth of surgical interventions and the steep rise in cancer cases, a pain crisis was in the making [74].

The development and introduction of a new formulation technology—sustained- or slow-release dosages that were characterized by releasing specific active drug compounds into the body over an extended period—changed pain medicine and the opioid concerns in medicine for the better (handling undertreatment) and for the worse (inducing overtreatment). In the 1970s, the Scottish drug producer Bard Laboratories introduced this technological innovation into the burgeoning field of pain medicine [75]. In 1980, the producer obtained an English license for selling a sustained-release preparation of morphine under the brand name MST [76]. MST was marketed as a revolutionary step in the transition of cancer pain treatment from pain relief to pain management. MST was initially considered a niche market product for palliative care within the growing field of end-of-life hospice care [77]. This would gradually change, however, after the take-over of Bard Laboratories by Purdue Pharmaceutical's English counterpart, Napp Pharmaceuticals. Through Napp Pharmaceuticals, Purdue obtained a license to develop the new preparation for the US market under the brand name MS-Contin.

The Reagan era of deregulation and continuing liberalization of the medical marketplace, with more liberal direct-to-consumer advertising regulations of drugs, created a fertile ground for post-marketing expansion of the MS-Contin medical market [78]. Without strong political support of the Reagan administration, the weakened and overburdened FDA regulators, faced with pressure from cancer patients and doctors mobilized by Purdue, ultimately accepted the novelty claim. They acknowledged the product claim that this

new drug formulation significantly reduced the risk of addiction and overdose in pain management with regard to morphine as a schedule II narcotic [30,79].

Purdue Pharma subsequently turned MS-Contin into one of its new cash cows within the growing market of oncological pain management. The financial support by Purdue Pharmaceutical of the leader in the field of oncological pain management, Russell Portenoy (from the world-renowned Memorial Sloan Kettering Cancer Center), proved paramount in mobilizing medical support for MS-Contin [80]. With his radical opioid-based pain management approach to chronic pain (both malignant and non-malignant), Portenoy took to the medical lecture circuit and published a series of articles [81,82]. He argued that, as long as patients had no history of drug abuse, the addictive risk of using opioids was very low [82]. His pain management movement gradually gained ground among medical opioid advocates, who engaged with a generation of American physicians with a low level of professional education about addictive substances [83,84].

This spurred Purdue management to develop a more potent pain medicine, one with morphine-like qualities but without morphine's phobic image, that could be used to achieve a significant growth in the market of chronic pain management [85]. Similar to Merck's German management's approach in the 1910s, the search was for an opiate-like compound that people would not associate with the stigma of morphine but that had similar therapeutic qualities. Oxycodone was lying idle on a shelf waiting to be invigorated again as part of a third drug life cycle as a follow up on Endo laboratories' successful Percodan and Percocet low-dose oxycodone aspirin/acetaminophen analgesic combination products. Though in the early 1970s the US government had classified oxycodone as a schedule II opiate drug, this did not prevent it from being 'rediscovered' by Purdue scientists. In imitation of MS-Contin, they reformulated oxycodone in the form of a slow-release pain medicine with proclaimed low addiction and overdose risks [86].

The FDA accepted the novelty claim without strong opposition and approved the new drug under the trade name Oxycontin® in 1995 for chronic cancer pain, thereby making it available as a 'scheduled narcotic' for prescription to all US doctors and their patients [87]. The FDA's European counterpart—the EMA—handled the Oxycontin registration dossier in a similar fashion and advised the European Commission to grant market approval of this narcotic drug for the indication chronic cancer pain without additional registration requirements [88]. Extra precautions such as strong addiction warnings in the medicine leaflet were not judged necessary by either regulator for introducing the narcotic on the medical market. Thus, while both the FDA and EMA fulfilled their duties in terms of product regulation, in terms of the also much-needed regulation and monitoring of marketing and health care practices, both agencies at the time did not have sufficient executive powers. It would last more than a decade before these shortcomings were repaired by the enactment in 2010 of the New EU Pharmacovigilance Legislation and in 2012 the FDA Safety and Innovation Act [89,90]. In the meantime, with insufficient pro-active vigilance of the regulatory systems, the potential for harm was significant [91,92].

However, it should be noted that the European Commission's product approval of Oxycontin did not automatically result in general availability and access in all European countries. National formularies have often been used in Western Europe to regulate what types of opioids may be prescribed under any set of circumstances [93,94]. Eastern European countries were particularly hesitant in making oxycontin and other new opioid drugs available and would impose additional regulatory restrictions to their medical use [95].

Purdue's marketeers would start challenging FDA and EMA's rather narrowly defined chronic cancer pain therapeutic drug indication for Oxycontin. In aiming at a swift expansion of the indication range towards the far more common non-cancer types of pain, they built on the legacy of the marketing genius and co-founder of Purdue—the American physician Arthur Sackler—who turned tranquillizers like Librium (chlordiazepoxide) and Valium (diazepam), known as benzodiazepines or 'benzos', into staples in any American medicine cabinet [76] (pp. 48–52). This involved skillfully promoting the new Oxycontin

drug directly to the community of health professionals through a number of means, including: glossy multipage color advertising in leading medical journals; publishing medical newspapers filled with promotional material and dubious paid-for scientific studies that exaggerated the problem, downplaying side effects and advocating new conditions for which the drugs would work; hiring thousands of doctors to promote the drugs (key opinion leaders or KOLs in the jargon); maintaining close relationships with FDA regulators; and monitoring doctors' prescription behaviors through the prescription-tracking company IMS (Intercontinental Medical Statistics) [96]. As such, Purdue's marketeers were able to take maximum advantage of the post-Reagan era of increasing entanglement of medical knowledge production and financial interests between drug companies, doctors and hospitals in the USA [97–99].

Imagery became as much a part of the fabric of Oxycontin's profiles as chemistry and pharmacology [100,101]. The circulation of the marketing driven images succeeded in bringing something immaterial to the drug itself: an aura of allure or fantasy, the mysterious fever of the benefit of the new, doing good and no harm to body and mind. All the lubricants for an Oxycontin-led pain medicine hype were available, including the direct-to-consumer advertising of a straight-forward fast-relief message that also fueled the antidepressant blockbuster era of the 1990s [102].

Moreover, the medical reputation of the previously lauded non-opiate NSAIDS analgesics had received a dramatic blow. The more people consumed NSAIDS, the more severe side effects (e.g., gastrointestinal bleeding, cardiovascular toxicity) became apparent and critical questions about the chronic use of this family of analgesics began to circulate first in medical quarters and later on in the public sphere [103]. This created a gap in the doctors' pain treatment armamentarium and whetted the appetite for better and smarter painkillers.

### *3.4. An Opioid Addiction Crisis in the Making: The First US Wave (1998–2010, see Figures 1 and 2)*

The closer ties in the 1990s in the USA between big pharma and the health care system played an important role in stimulating Oxycontin and other 'new' opioid consumption [104–108]. Alongside the fast-growing movement of pain treatment patient advocates, Purdue marketers portrayed pain medicine as a backward area in medicine that needed a radical change, not only in cancer pain centers but also in general practice and for chronic pain as well [109,110]. Purdue pharma and other opioid producers in conjunction with medical journals supported post-academic courses that paid increasing attention to the recognition, diagnosis and treatment of pain. The idea was planted that doctors under-treated pain, and that the treatment of pain was a 'fundamental human right and duty' [111]. Prescription monitoring of narcotic drugs was being portrayed as frustrating the efforts to modernize pain medicine and undermining the autonomy of doctors. The overall message was that pain must be treated, preferably with a new generation of slow-release opioid drugs with a negligible risk of iatrogenic addiction [112]. Perhaps the most cited case in point is the uncritical appraisal and misrepresentation in marketing practices of a 1980 study in the prestigious *New England Journal of Medicine* that claimed that the development of opioid addiction is rare in hospitalized medical patients with no history of addiction [113–115].

However important the impact of this aggressive kind of pain management mongering might have been, it still depended on physician and consumer support and regulatory failure to be successful. In the decade of public optimism about uncovering the secrets of life in both the human genome and the brain, the promise of fast pain relief was embraced in the consultation room [116]. This resulted in pain being included as a fifth vital clinical sign that needed treatment. The American Pain Society and the American Academy of Pain Medicine issued a consensus statement endorsing opioid use for chronic non-cancer pain [117]. State medical boards and state policies started to relax regulations about prescribing opioids to non-cancer patients [118]. In support of these relaxing opioid prescription policies, the American Medical Association seemed to be too optimistic about

physicians' professional role as opioid gatekeepers [104] (pp. 62–64). In addition, the Center for Medicare and Medicaid Services (CMS) incorporated pain management into patient satisfaction scores, thereby linking patient experience and pain management to reimbursement. CMS reimbursed physicians for their services based on prescription value, incentivizing high-value long-term prescriptions. Long-term use of slow-release opioids became a financially rewarding opportunity for US prescribers [119,120].

Furthermore, changing quality standards for hospital care and the rise of polyclinic or outpatient surgery, in combination with the professional support of pharmacological pain treatment programs and the existence of unmonitored opioid pharmacies (so-called pill mills), have also played an important role in the growth in consumption of opioid painkillers [121]. Hospitals are required to measure their quality of care and pain is one of the indicators that determines quality. The results of these quality measurements are made public, and hospitals advertise these figures for marketing purposes. The less pain that patients experience, the higher the hospital's rating [122]. These pro-opioid institutional and regulatory shifts helped to further open the gates for massive overprescription of Oxycontin in the USA—a combined pain and opioid epidemic was in the making (See Figures 1 and 2).

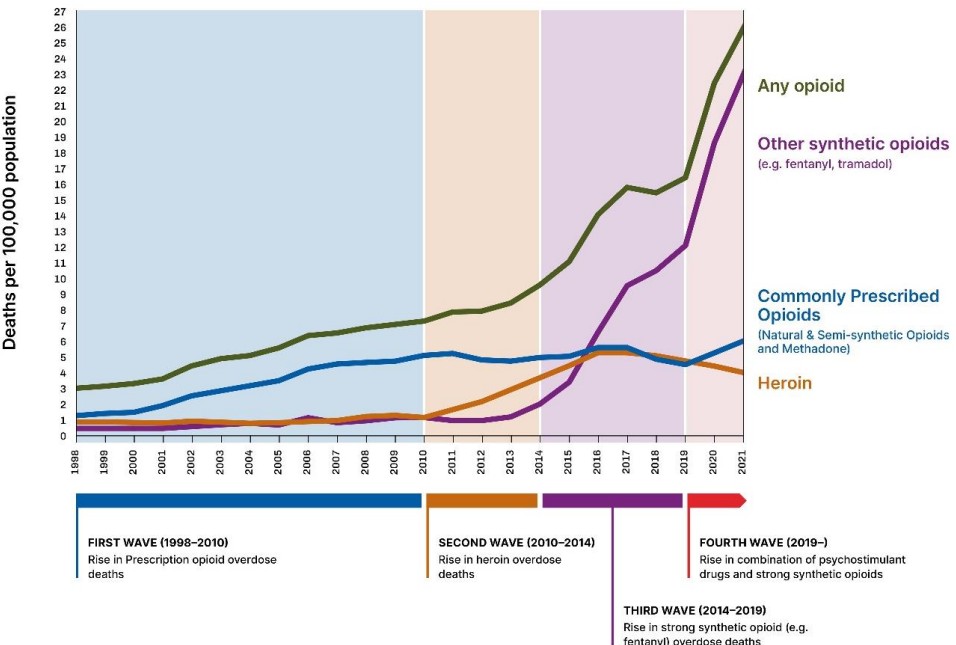

**Figure 1.** Redrawn and modified from CDC figure [123].

Growing focus on pain management meant growing opioid sales, with direct-to-consumer ads by Purdue Pharma, like 'Oxycontin: It gets you high' or 'I couldn't get through the day without it' in the USA [124]. There was a definite racial subtext to the Oxycontin ads, which targeted primarily Caucasian white consumer groups that were not thought to be at risk of addiction (i.e., suburban and rural non-Hispanic white populations). Counties with a higher degree of rurality appeared to have higher opioid prescribing rates and this association could be explained by higher percentages of whites, higher unemployment rates, less nurse practitioners and physician assistants and more specialized opioid prescribers such as surgeons and oncologists [125]. At the same time, racial prejudice and lower health insurance coverage rates protected African American and Hispanic communities, associated with the heroin epidemic of the 1970s, against overprescription [60] (p. 151).

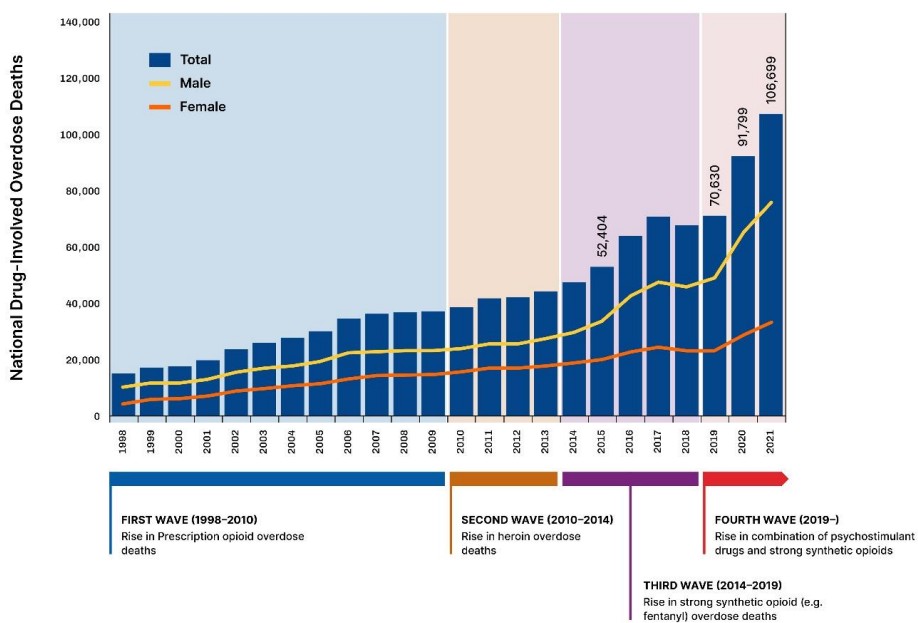

**Figure 2.** Redrawn and modified from CDC figure [123].

Hospitals and health insurance plans were largely in support of the new pain management programs that included chronic pain management (e.g., for musculoskeletal pain problems), in addition to acute pain (cancer and post-operative). Opioids were indiscriminately promoted for both types of pain management, despite lack of evidence of the effectiveness of prolonged opioid use in the case of the most frequent chronic non-cancer pain [126]. Between 1999 and 2010, the medical sales of opioids in the USA quadrupled. Following Purdue Pharma, other drug companies like Endo Pharmaceuticals (DuPont), Abbott, Johnson and Johnson's Jansen Division and the generics companies Mallinckrodt pharmaceutical and Teva's Malvern-based Cephalon unit jumped on the slow-release opioid bandwagon with oxymorphone (Opana®, a Normorphan make-over), hydromorphone (Dilaudid®), tapendatol (Nucynta), oxycodone (Roxicodone®) and fentanyl (Duragesic®, Actiq® lollipop, Fentora®) [127].

All these companies, Purdue Pharma included, used the opium alkaloid thebaine as the key chemical precursor for the production of semi-synthetic opiates, except for fentanyl and tapendatol which are synthetic opioids. Around the early 1990s, plant biologists at the Tasmanian Alkaloids facility (Johnson and Johnson subsidiary), together with professor Meinhardt Zenk, succeeded in genetically modifying the opium poppy Papaver somniferum to produce a morphine-free poppy plant variety containing high thebaine concentrations [128]. This catapulted the highlands of Tasmania into the nucleus of the global opioid supply chain, with a more than 50% share of the global thebaine supply [129]. Research has shown the influence that pharmaceutical lobbyists had on more relaxed control regulations for thebaine than in the case of the conventional licit opiate raw materials from Turkey and India [130]. The regulatory loophole of thebaine production and supply was used by Johnson and Johnson to meet the global demands for raw thebaine material by all major opioid producers, until the Tasmanian Alkaloids facility was sold in 2016 [131]. By that time, the international relocation of pharmaceutical ingredients and chemical precursors production to the low-cost countries of India and China would further fuel the international licit and illicit opioid production and consumption.

Oxycontin jumpstarted the US opioid epidemic, which would subsequently evolve as a series of four intertwined but distinct epidemics—with a variety of opioids over the four epidemics associated with mortality and with diverse geographical, temporal and sociodemographic patterns. In the first wave, opioid-related deaths were mainly associated with prescription opioids such as Oxycontin (oxycodone hydrochloride), starting in 1998

until approximately 2010. The hardest-hit US communities during the first wave were found in the east-coast states of West Virginia, Ohio, Kentucky and New Hampshire. These communities were most affected by economic decline due to the massive loss of industry-based jobs in the USA in the 1990s. The decline caused a lot of pain and despair-related problems over the years—from unemployment to broken families to poor health [132,133]. In addition, these states also have a high prevalence for incapacity for work among the population due to long-term heavy physical labor and a high prevalence of chronic pain. Furthermore, these aforementioned states had no state-controlled prescription monitoring programs to prevent the overprescription of opioids and other schedule II drugs [134].

*3.5. The Second (2010–2014), Third and Fourth Wave of the Opioid Crisis (See Figures 1 and 2)*

From 2010 onwards, the second wave was associated with rapid increases in heroin-related overdose deaths and a shift youthwards. This coincided with the increased controlling and monitoring of the legal access to prescription opioids and the introduction of an abuse-deterrent formulation (ADF) of OxyContin by Purdue Pharma, that made it difficult to abuse the drug in this fashion [135]. Criminal entrepreneurs and networks fulfilled the needs of large numbers of prescription-opioid-addicted patients, who either no longer had access to medical market sources due to more stringent prescription regulations, had insurance challenges or were dissatisfied with the new abuse resistant oxycodone pills. Mexican transnational criminal organizations, which from the 1990s controlled most of the US illegal drug markets, sensed a fast-emerging new market in the USA [136,137]. They began to aggressively supply massive amounts of cheap heroin to partner criminal groups and gangs in the United States. American consumers by the hundreds of thousands resorted to these non-medical black-market opiate channels. Heroin became widely regarded by iatrogenic-opioid-dependent US users as a suitable and cheap black-market alternative for the opioid pain relievers [138].

In the third (2014–2019) and current fourth wave (2019–) of the crisis (see Figures 1 and 2), the further increase of opioid use and overdose death rates were associated with Chinese-manufactured, cheap and extra-strong synthetic opioids such as fentanyl and fentanyl analogs (e.g., the most dangerous fentanyl derivative is carfentanyl) as well as the combination of psychostimulant drugs and opioids, which are primarily distributed via the existing non-medical criminal channels [139,140]. During the third and fourth waves, the opioid epidemic spread quickly to US communities both on the east and west coast that were initially hardly affected, including African American, Hispanic and other ethnic minority groups [132,141]. Existing inequalities within society related to socioeconomic status and race have become increasingly important in the third and fourth wave of the opioid crisis. Three clear factors for overdose deaths are currently: coming from a deprived background, being from a racial or ethnic minority group and being part of the 1981–2000 millennial generation [142,143].

National and local efforts to deal with the opioid epidemic have been on the rise. Possession of illegal drugs and the illicit use of legal drugs are still US federal crimes and prisons are still filled with convicts convicted of these crimes [144]. But, according to the former editor-in-chief of the New England Journal of Medicine Marcia Angell, many states, counties and cities in the USA have begun to regard opioid addiction more as an issue of public health than of criminal justice [79]. As part of a new harm reduction approach, centers are being opened in which people who seek help are treated with opioid replacement therapy (e.g., methadone and buprenorphine (Subutex)— a method known as 'medication-assisted treatment' or MAT). Naloxone (Narcan), the antidote for an opioid overdose (opium antagonist), is now sold over the counter as an emergency kit in almost all US states. If used immediately and properly, it can prevent an otherwise inevitable death from a drug overdose. It is also becoming more common for some drug courts to drop criminal charges in return for an agreement to submit to treatment and monitoring [79]. At the same time, US pharmaceutical companies and drug wholesalers have been on trial for helping to fuel the deadly opioid crisis. This has resulted in more than 30 billion of

dollar federal opioid settlements [145,146]. But the money for devastated communities will arrive at the moment when the opioid crisis has escalated dangerously, and it proves rather difficult to close the illicit opioid gates.

The volatile politics of drug regulation both nationally and internationally continue to mold the US opioid crisis that started as an iatrogenic epidemic but developed into an illicit-opioid-fueled crisis that is currently dominated by trafficking fentanyl and fentanyl analogs that are primarily produced in Asia. According to Brookings Institute researcher Vanda Felbab-Brown, there is no easy way out due to the lack of 'global political appetite for scheduling a vast number of dual-use (medical/non-medical) chemicals', and the rather tense bilateral relationship between the US and China is not of any help [147,148].

The opioid epidemic is not only rampant in the United States. Canada is facing a similar crisis, driven by both prescription and illegal opioid use, with significant overall increases in opioid-related deaths and a marked increase in fentanyl-related deaths in some provinces [149]. Prescription opioid use also appears to be an early driver of the Canadian crisis, while the increasing availability of opiates and opioids on the illegal market is likely driving the most recent rise in deaths in most Canadian jurisdictions [150]. In addition, on the other side of the Atlantic, there are also signs of an emerging iatrogenic driven epidemic, albeit in a more silent and nuanced fashion.

In Europe, the medical use of opioids has substantially increased since 2009 [151]. However, the situation in Europe differs significantly from the USA and Canada. Specifically, the regulatory and health insurance contexts and approaches to opioid medicine marketing and prescribing are rather different in Europe. In Europe, direct-to-consumer advertising by pharmaceutical companies is not allowed, so-called pill mills do not exist and most citizens have adequate health insurance coverage and feel no need to search for alternative sources of narcotic drugs. Furthermore, most European countries have vertically integrated healthcare systems facilitating effective control of opioid prescribing. National and local formularies play an important role in restricting availability and limiting the circumstances under which opiates can be used. Moreover, there are multiple legal barriers to accessibility. All of the East European countries and some of the West European countries (e.g., France, Greece, Portugal) require that opioids be prescribed using duplicate or triplicate prescriptions. In most of these countries, special forms must be used which, in some cases, physicians need to purchase [152,153]. Finally, the European health care systems are organized differently with a different reimbursement structure, patient satisfaction monitoring and pharmaceutical price regulation [154]. Together, this leads to more administrative burden and less incentive to prescribe opioids than in the US, resulting in less opioid use.

In addition to these limiting factors, the warning example of the enormous opioid epidemic in the USA has so far prevented a similar opioid epidemic from happening in Europe. As of yet, there has been no alarming increase in the number of opioid-related deaths and only recently a rise can be seen in the number of patients in addiction treatment for opioid use disorders [155]. Still, the number of opioid prescriptions throughout Europe has continued to rise during and after the COVID-19 lockdown period and further close monitoring is required to prevent epidemic levels from being reached, all the while ensuring adequate pain control for patients [156,157]. The opioid epidemic has possibly also spread to parts of Africa, Asia and South America, but those problem areas will not be addressed here [158–160].

## 4. Discussion and Conclusions

Medical gatekeeping of prescription opiates and other psychoactive drugs was a product of the American and European opiate epidemic in the late nineteenth and early twentieth centuries. Doctors and pharmacists were officially acknowledged as gatekeepers, but neither they nor governments on both sides of the Atlantic anticipated possible major changes in the supply and demand dynamics of narcotic drugs. All parties were overwhelmed by the consequences of the imperative of drug regulation and commercially driven pharma-

ceutical innovation, with an ever-expanding list of controlled psychotropic prescription drugs that required monitoring by trained medical professionals. The prescription-only regulations of psychotropics, however, did neither prevent the phenomenon of addiction on prescription nor the emergence of a non-medical market. Instead of hard boundaries between medical and non-medical production, boundary crossing of distribution and consumption channels became the rule rather than the exception in the post-World War II period. The utopian notion of a 'pill for every ill' perfused the capillaries of society. The pharmaceuticalization of everyday life took a new pervasive turn in the 1990s in the US, with its uniquely unbridled forms of direct-to-consumer marketing of therapeutic drugs and the consequential tripling of US prescribed drug sales between 1980 and 2000 [161]. The pharmaceutical industry and its lobbying forces pervaded medicine and the social fabric of US society [162]. There is consensus that the pill-pushing pharmaceutical industry, together with the rise of a corporate healthcare system that is run like a business, is what spawned the US opioid epidemic.

American doctors are for the greater part in private practice and benefit financially from the number of patients they treat and prescriptions they write. This factor has certainly incentivized the overprescription of pain medication, but the aggressive marketing of medication-centered pain-management programs by opioid manufacturers (who often paid for training doctors and pharmacists) primarily contributed to the soaring appetite for quick-relief pain medication. Ultimately, the medical and non-medical drug production, distribution and consumption channels conflated into a self-perpetuating contradiction and delusion of control. Existing socioeconomic inequalities within US society contributed to the unprecedented persistent nature of the opioid crisis with four consecutive waves.

In retrospect, regulatory bodies could have been more vigilant from the start by performing similar kinds of thorough, long-term life-cycle analyses of oxycodone—with three consecutive drug life cycles—and other controlled drugs, as shown in this article. The patent on oxycodone was renewed after each reformulation, with claims about improvements (e.g., adding wax would make it less crushable, so it could not be snorted, and the patent was renewed, thus ensuring exclusivity and profitability was guaranteed for years). Without the loopholes in the regulatory system, Purdue Pharma could never have profited for so long. Furthermore, without additional effective post-marketing regulatory surveillance there was no way 'to prevent catastrophe from being the first evidence of a previously unsuspected major hazard in a marketed drug', as the regulatory expert John Urquhart eloquently phrased it [163]. The failure of the US regulatory authorities to effectively respond to the 'corporate-made' opioid crisis was evidently due to the focus gap between product regulation and regulation of marketing practices, as well as due to under-regulated health care practices. Hopefully, the 2012 FDA Safety and Innovation Act will prevent this from happening in the future.

As professional opiate gatekeepers, US doctors and pharmacists did not sufficiently prevent the opening of the narcotic gates they were supposed to guard for the common good, informed by pharmaceutical companies with biased data and advertising. Once the gates were opened, medical professionals were hardly able to curb the forces of habit [164,165]. Ironically, this turned the most powerful geopolitical force in the war against drugs into its greatest victim. Due to formulary limitations, regulatory barriers to accessibility and differences in financial structures in health care, European countries have been relatively protected against following the suit of the US and Canadian opioid crisis. But they should also be aware that the spiraling of medically controlled licit ('on prescription') drug markets and criminally controlled illicit drug markets might have their own uncontrolled future dynamics.

In a first effort to regain control (and more effectively prevent iatrogenic addiction), doctors and pharmacists, supported by regulatory authorities, could develop appropriate training programs and safe-guarding professional independence [166]. This may also help to prevent a new era of opiophobia and a painful underprescription of opioids. Human Rights Watch reported in 2018 that the current debate on opioid overdosing has

led American physicians to cut back on or completely withhold opioids for patients in acute pain or chronic cancer pain, which may force those in need of pain relief to resort to non-medical sources—a gap eagerly filled by criminal entrepreneurs, as we have seen in the second, third and fourth waves of the US opioid epidemic [167]. In addition, it is advisable to curb patent rights on reformulations and reconsider financial prescription incentives.

Providing patients and prescribers with the tools to combat pain while limiting the potential for abuse is a difficult balance to strike. Comparisons between different countries' regulatory and market access environment, prescriber and distributor experiences and their implications for patients allow us to improve and prevent both over- and underprescription of opiates. All stakeholders—regulators, prescribers, pharmacists and pharmaceutical companies—have a responsibility to improve on prior practice based on the lessons learned in the several waves of opioid crises.

Given the narrative-based review approach in this article, with all its methodological caveats of presenting a view or approach that may be biased and difficulty in ascertaining the completeness and representativeness of the literature presentation, it is important that further systematic studies are undertaken to develop evidence-based implementation strategies for providing opioid pain medicine for the benefit of individual patients. The generalizations in this article do not imply the individual responsibilities of doctors, pharmacists and regulators.

**Funding:** This research received no external funding.

**Acknowledgments:** The author would like to thank Stephen Snelders and Toon van der Gronde for their feedback on this paper. Furthermore, I would like to thank Kathryn Burns and Cassandra Nemzoff for their English manuscript correction services. In addition, I would like to thank Frank-Jan van Lunteren for his comprehensive Figures 1 and 2 graphics. This article was partly written during the NWO project 'The Imperative of regulation' (project nr. 360-52-180, 2016–2021).

**Conflicts of Interest:** The author declares no conflict of interest.

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
