# Peer review of "The Imperative of Regulation: The Co-Creation of a Medical and Non-Medical US Opioid Crisis"

_psychoactives, doi:10.3390/psychoactives2040020_

Round 1
Reviewer 1 Report
Comments and Suggestions for Authors
Overall review: This is a really interesting and important article; its argument is highly complex, yet concise and highly readable. The article can be published as it is, I just have a few minor suggestions (see below).
The paper points to the link between the medical system, the pharmaceutical market and an increasingly market-driven health care system; it employs a life-cycle approach to the opioid crisis. The paper addresses the different waves of the opioid crisis, taking into account doctors’ gatekeeping function as well as the role of the FDA and the war on drugs, political regimes.
Method: Literature review of Dutch and German language publications.
Optional suggestions: It would be interesting to note possible differences between the US and Europe, where there is a less market-driven approach to health care. It would also be interesting to learn a little bit more about the life cycle approach and its link to specific disciplines, e.g., sociology or Science and Technology studies.
Author Response
Dear reviewer,
Thank you for your positive review. As far as the materials and method section is concerned I have added a reference to JP Gaudilliere (drug trajectories) and in the text I added the fact that we built on the work in the field of science and technology studies in which I participated within the context of the Maastricht Bijker group. The optional suggestions will be addressed in a follow-up paper that I am currently working on.
with kind regards,
Toine Pieters
Reviewer 2 Report
Comments and Suggestions for Authors
The review manuscript, presented in a narrative style, delves into the pivotal occurrences that gave rise to the opioid crisis in the United States. Drawing from referenced sources, the manuscript illustrates how pharmaceutical companies involved in opioid production played a significant role in establishing a medical market for opioid painkillers and stimulating consumer demand. Moreover, it highlights the inadequacies of US regulatory bodies, notably the FDA, in addressing this crisis effectively, attributing this failure to the disconnect between narcotic product oversight, marketing practice regulation, and the emergence of a healthcare system dominated by corporations. As a result, the manuscript elucidates how these factors have culminated in the ongoing opioid epidemic.
In general, this review is thoughtfully organized and effectively presents a timeline of significant events, making it appealing to a broad readership. Given the abundance of recent publications focusing on the US opioid crisis and its connection to inadequate primary prevention, it would have enhanced its value to readers if the author had included a more comprehensive discussion of the latest findings regarding strategies to combat the opioid epidemic and the advantages and drawbacks of prescribing controlled substances.
Author has used relevant references and cited them appropriately through the manuscript.
I do not have any major concerns or suggestions.
Author Response
Dear reviewer,
Thank you for your positive review. I acknowledge the fact that I could have included latest findings regarding strategies to combat the opioid epidemic and the advantages and drawbacks of prescribing controlled substances. I am currently working on a follow-up paper that will address these issues.
with kind regards,
Toine Pieters
Reviewer 3 Report
Comments and Suggestions for Authors
Reference number: Psychoactives-2660031
Authors: Pieters Tonie
Title: The Imperative of Regulation: The-Co-creation of a Medical and Nonmedical US Opioid Crisis
In this narrative review the author provides us with a historical overview of the US Opioid Crisis, presents aspects discussed in literature and highlights some of its mechanisms based on the drug life cycle of oxycodone. The author further discusses the possible causes and attempts to identify the most likely switching points to gain control over this crisis, but also admits, given the approach of the review, the danger of biases and thus the need of a systematic approach.
This is an important discussion but also a difficult one. Given the complex nature of the crisis it is easy to get caught in bias or miss aspects that change the context. The urge for action can lead us to hastly accusations.
The properties of opium have been known since antiquity and the application of morphine and other narcotic analgesics will also be required in the future. The pharmaceutical companies and also generations of researchers working in this field invested a huge amount of hard work and time to discover the optimal/ideal analgesics without any side effect. It is noticeable, that this perspective is missing in the review which raises the worries that we might create a new cyclic pattern: The same way an overall positivism contributed to the blanking out the risks of opioid treatment, an overall negativism given the known result of a crisis could make us forget the benefits and success.
This would transfer the discussion of risk management and of responsibility/causality in complex systems (society) into a blame game and mask the underlying pattern: the failure of abstraction.
This manuscript is enjoyable and well-written. I strongly recommend this manuscript for publication after revision of some minor issues.
Specific comments
I would like recommend the depiction of the basic chemical structures which are subject of this work: morphine, heroin, cocaine, oxycodone, oxymorphone, hydromorphone, fentanyl, methadone, naloxone, buprenorphine (i.e., two possible versions of figure attached).
Page 10 Line 464–466
“All these companies, Purdue Pharma included, used the opium alkaloid thebaine as the key chemical precursor for the production of opioids, except for fentanyl and tapendatol which are synthesized differently.”
All these companies, Purdue Pharma included, used the opium alkaloid thebaine as the key chemical precursor/starting material for the production of semi-synthetic opiates, except for fentanyl and tapendatol which are synthetic opioids.
Page 10 Line 466–469
“Around the early 1990s, plant biologists at the Tasmanian Alkaloids facility, a subsidiary of Johnson & Johnson, succeeded in genetically modifying the related opium poppy Papaver bracteatum to produce a variety containing high thebaine concentrations.”
“Papaver somniferum”
It has been known since the 1970s, that thebaine predominates and only small amount of other alkaloids are present in Papaver bracteatum (Nyman, U.; Bruhn, J. G. Papaver bracteatum – a summary of current knowledge. Planta Medica 1979, 35, 2, 97–117).
It is necessary to emphasize the paramount importance of the discovery of the researchers of the Tasmanian Alkaloids together with professor Meinhardt H. Zenk which made available the morphine-free poppy plant (Millgate, A. G.; Pogson, B. J.; Wilson, I. W.; Kutchan, T. M.; Zenk, M. H.; Gerlach, W. L.; Fist, A. J.; Larkin, P. J. Morphine-pathway block in top1 poppies. Nature, 2004, 431, 413–414). The genetically modified Papaver somniferum (TOP = thebaine – oripavine – poppy) contains predominantly thebaine and oripavine (morphine: 0 %, codeine: 0 %, oripavine: 0.43 %, thebaine: 1.65 %). Thebaine itself was regarded for a long time as a useless side-product of the poppy extraction. It is a toxic alkaloid, which causes spasms/cramps and has no analgesic effect. On the other hand, the industrial procedures for conversion of thebaine to pharmacologically useful APIs were well known since a long time: morphinan-6-ones [14-hydroxydihydrocodeinone (oxycodone), 14-hydroxydihydromorphinone (oxycodone)], “Nal-compounds”: naloxone, naltrexone, nalbuphine, 6,14-ethemomorphinans (buprenorphine, etorphine, diprenorphine). In this way it became possible to prevent the illegal access to morphine and codeine containing poppy plant/straw because the new TOP1 Papaver somniferum was useless as raw material for illegal manufacturers or as narcotic drug.
Page 11 Line 512–513
“and extra-strong synthetic opioids such as fentanyl and fentanyl precursors as well as the combination of”
Use fentanyl analogues, fentanyl related compounds or fentanyl derivatives instead of “fentanyl precursors” please.
In this section consider mentioning one of the most dangerous fentanyl derivative: carfentanil (see e.g., Shafer, S. L. Carfentanil: a weapon of mass destruction. Can. J. Anesth. 2019, 66, 4, 351–355.)
Page 12 Line 542–543
“is currently dominated by trafficking fentanyl and precursor agents that are primarily produced in Asia”
Use fentanyl analogues, fentanyl related compounds or fentanyl derivatives instead of “precursor agents” please
References
Text from EndNote (or from another literature managing software) was imported incorrectly in some cases. Please correct. Mostly German characters/fonts (Ü, ü, ä, )
Page 17 Line 776
[54] Über…. Zeitschrift für Angewandte
Page 17 Line 779
[57] und Bekämpfung
Page 17 Line 782
[58] Menninger, E., Bachem, C. Eukodal-Vergiftung, chronische. (Eukodalismus.). Sammlung von Vergiftungsfällen 1932, 3, 173–174. Doi: 10.1007/BF02455131
[61] ge?ist tegen
[66] J¨¹nger, S.,

Author Response
Dear reviewer,
Thank you for your kind words. It is indeed a rather complex crisis and I fully acknowledge the fact that more systematic studies should be done, hopefully as soon as possible.
I have revised the manuscript in line with your minor edit remarks. The German spelling was different in my manuscript, but the check made me aware that the references needed yet another check.
with kind regards,
Toine Pieters